# Potential of Heterogeneous Compounds as Antidepressants: A Narrative Review

**DOI:** 10.3390/ijms232213776

**Published:** 2022-11-09

**Authors:** Gonghui Hu, Meng Zhang, Yuyang Wang, Ming Yu, Yu Zhou

**Affiliations:** 1Department of Rehabilitation Medicine, Affiliated Hospital of Qingdao University, Qingdao 266000, China; 2Department of Physiology and Pathophysiology, School of Basic Medical Sciences, Qingdao University, Qingdao 266071, China; 3Institute of Brain Sciences and Related Disorders, Qingdao University, Qingdao 266071, China; 4Neuroscience and Neurorehabilitation Institute, University of Health and Rehabilitation Sciences, Qingdao 266000, China

**Keywords:** depression, antidepressants, endogenous compound, review

## Abstract

Depression is a globally widespread disorder caused by a complicated interplay of social, psychological, and biological factors. Approximately 280 million people are suffering from depression worldwide. Traditional frontline antidepressants targeting monoamine neurotransmitters show unsatisfactory effects. The development and application of novel antidepressants for dissimilar targets are on the agenda. This review characterizes the antidepressant effects of multiple endogenous compounds and/or their targets to provide new insight into the working mechanism of antidepressants. We also discuss perspectives and challenges for the generation of novel antidepressants.

## 1. Introduction

Depression is a prevalent and multifactorial mood disorder. The neurobiological basis of depression is poorly understood. Conventional frontline antidepressants, including selective serotonin reuptake inhibitors (SSRIs), norepinephrine reuptake inhibitors (NRIs), and serotonin and norepinephrine reuptake inhibitors (SSNRIs), generally exhibit delayed action and have multiple adverse effects [1]. Due to a large social need, there is a burgeoning interest in developing novel antidepressants. For example, ketamine (a rapid-acting antidepressant) alleviated depressive symptoms within hours of administration in contrast to monoamine antidepressants [2].

The traditional “serotonin hypothesis” indicates that a deficiency of serotonin exacerbates depressive symptoms [3]. However, a large body of studies showed that abnormal monoamine transmitters/receptor systems, altered stress hormone dynamics, deficient growth factors and neurotrophins, dysregulated pro-inflammatory cytokines, disturbed adult neurogenesis, altered synaptic connectivity, oxidative stress, abnormal miRNA expression, and abnormal delivery of gastrointestinal signaling peptides are associated with the pathogenesis of depression [4,5,6,7]. Specifically, recent studies demonstrated key factors associated with depression, such as chronic stress, that lead to increased glutamate levels and excessive activity of N-methyl-D-aspartate (NMDA) receptors. Imbalanced glutamate neurotransmission eventually causes excitotoxicity and leads to the atrophy of emotional neurons [8]. Ketamine is an NMDA antagonist with a rapid onset for the treatment of the major depressive disorder (MDD) with suicidality. However, it is discouraging that the efficacy of ketamine in acute patients is not sustainable, and most patients do not achieve adequate alleviation [9]. Therefore, the determination and development of alternative rapid-acting antidepressants are needed. In this review, we first provide a brief overview of the neuropathophysiological alterations in depressive patients. We then focus on introducing recent studies on endogenous compounds and their receptors that exhibit antidepressant properties to discuss innovative ideas for the development of novel antidepressants.

## 2. Neurological Alterations in Depression

Critical brain regions associated with depression include the medial prefrontal cortex (mPFC), the hippocampus, the amygdala, the dorsal raphe nucleus (DRN), the lateral habenula (LHb), the anterior cingulate cortex (ACC), the nucleus accumbens (NAc), and the ventral tegmental area (VTA) [10]. The mPFC has been the focus of research to understand structure–function relationships in depression. The hippocampus is one of the key brain structures that show morphological changes associated with depression and stress. The amygdala modulates emotional and affective aspects of cognition. The DRN is the largest 5-hydroxytryptaminergic nucleus sending divergent projections to many functionally differentiated areas that control motivation, arousal, sleep, and mood. The LHb handles negative valence information and reward prediction errors. The ACC is part of the affective, sensory, and cognitive systems that govern motivational endowment, social self, and personality. The NAc is a robust motivation-promoting center that integrates signals encoding learning, memory, emotion, and evaluation. Dopaminergic neurons in the VTA are closely correlated with the reward learning, motivation, and anhedonic features of depression. All of these regions may undergo defects [10], such as volume, neuroplasticity, neural connectivity, synapse formation, and neurogenesis.

The volume of the subgenual PFC exhibited an approximately 40% decrease in depressive patients, with reduced neuroplasticity and smaller neurons [11]. The diminished function of the subgenual PFC activated the hypothalamic–pituitary–adrenal (HPA) axis and sympathetic nervous system, which suppressed the activation of the nucleus accumbens (NAc) and caused anhedonia in depression [10]. The gray matter of the dorsolateral PFC was also decreased in depressed patients [11]. Notably, neuroplasticity in the amygdala was increased [12], and the activated amygdala contributed to the hyperactivity of the HPA axis and sympathetic nervous system in depressive patients [10]. The hippocampal volume was reduced, and neuroplasticity and neurogenesis were significantly diminished [11]. In contrast, the volume of the NAc was increased, and the response to pleasurable stimuli was diminished [11]. The habenula exhibited decreased volume and increased burst firing that inhibited the release of dopamine from the ventral tegmental area (VTA) to the NAc, which resulted in anhedonia [11]. In addition to altered site structure and function, depression is correlated with synaptic loss and functional connectivity defects [13]. Stress and inflammation are inextricably linked to depression, and the brains of depressed patients are in a markedly proinflammatory state. Microglia activation, cytokines, corticotropin-releasing hormone (CRH), and other inflammatory mediators contribute to neuroinflammation in the central nervous system (CNS) [14].

Alterations in the endocrine system affect the onset and development of depression. For example, CRH is hypersecreted in depression [11], and insulin resistance is prevalent in depressive patients [15]. Abnormal glucocorticoids cause a decreased volume and diminished neuroplasticity in the subgenual PFC and the hippocampus and contribute to amygdala enlargement and increased neuroplasticity in this structure [11]. Marginal salt corticosteroid receptors (MRs) are downregulated during chronic stress and depression [16], and estrogen receptor (ER) β agonists increase the expression of critical synaptic proteins, including postsynaptic density 95 (PSD-95), synaptophysin, and α-amino-3-hydroxy-5-methyl-4-isoxazole-propionic acid (AMPA) receptor subunit glutamate receptor 1 (GluR1) [11]. Hippocampal volume is decreased in patients with hypothyroidism [17]. CRH or glucocorticoid receptor antagonists, insulin receptor or MR agonists, allopregnanolone, and thyroid axis drugs may be new treatment modalities against depression.

## 3. New Hotspots in Antidepressant Research

Conventional antidepressant treatments are inadequate for the disastrous consequences of depression. The exploitation of novel antidepressant drugs and the minimization of the adverse effects of current antidepressant drugs will greatly benefit patients and society. Although the serotonin transporter (SERT) is presently the primary target for antidepressants, recent research highlights the fact that some endogenous neuropeptides, hormones, and lipid neuromodulators may act directly or indirectly to alleviate mood disorders, suggesting that targeting these molecules or compounds and/or their receptors may eventually help to develop new strategies to treat depression. Therefore, in this review, we focus on current findings regarding the modulatory effects of histamine, acetylcholine, peptides, and lipids on depression.

### 3.1. Histamine

Brain histaminergic systems play a critical role in cognition, sleep, and neuropsychiatric disorders, including Alzheimer’s disease (AD) and Parkinson’s disease (PD) [18]. The prominent engagement of brain histamine (HA) in brain disorders demonstrates that HA and receptor systems are invaluable areas for studying the etiology of associated brain disorders. HA is primarily derived from histaminergic neurons expressing histidine decarboxylase (HDC), which are located solely in the tuberomammillary nucleus in the posterior hypothalamus and project axons throughout the brain [19]. HA exerts its function by mobilizing four G protein-coupled receptors (H1R, H2R, H3R, and H4R) [20]. Several reports demonstrated that histamine modulates the proliferation and differentiation of neural stem cells and contributes to synaptic plasticity during embryonic and postnatal development [21].

H3R is the most abundant histamine receptor in the CNS. A presynaptic inhibitory H3 self-receptor regulates HA synthesis and release in a negative feedback mechanism. H3R is also localized on non-histaminergic neurons and regulates the release of many other neurotransmitters [22]. H3R-mediated HA release interacts with postsynaptic H1R and H2R or indirectly modulates cholinergic, dopaminergic, and γ-aminobutyric acid (GABA)-ergic neurotransmitters [23]. This phenomenon renders agents, such as H3R antagonists/inverse agonists, attractive therapeutic targets for CNS disorders. The older generation of the H3R antagonist/inverse agonist clobenpropit (Figure 1) had an antidepressant-like effect in rats [24]. The H3R antagonist/inverse agonist JNJ-10181457 exhibited antidepressant-like effects in lipopolysaccharide (LPS)-induced depression model mice and suppressed proinflammatory cytokine release from microglia [25]. Decreased brain-derived neurotrophic factor (BDNF) in the hippocampus and PFC of mice exposed to chronic unpredictable stress was reversed by the administration of ciproxifan (H3R antagonist/inverse agonist) in preclinical depression models [26]. Substantial efforts have been devoted to explore new drugs that specifically target histamine H3R for the treatment of depression and other cognitive disorders. ST-2300 is a novel multi-active ligand with an affinity for H3R and a 5-HT2A that exhibited antidepressant and anxiolytic-like effects in animal models of depression [27]. Therefore, ST-2300 may be a potential treatment for depression and anxiety-related neuropsychiatric disorders.

The H1R antagonist clemastine diminished proinflammatory cytokines and reversed depression-like behavior in a chronic unpredictable mild stress (CUMS)-induced mouse model of depression [28]. JNJ7777120 (H4R antagonist) diminished glutamate-dependent neurotoxicity by upregulating glutamate transporter expression and increasing glutamate reuptake [29]. JNJ7777120 completely prevented the impact of HA on BDNF in primary cultured neurons, which indicates that H4R may play a role in restoring BDNF. This result is pertinent to the findings that H4R knockout mice exhibit depression-like symptoms [30]. The exploitation of antagonists or inverse agonists targeting histamine receptors has unpredictable potential for the treatment of depression.

### 3.2. Acetylcholine

Acetylcholine (ACh) is a potent modulator of neuronal activity in the CNS. Although the predominant drugs for MDD are currently geared toward the monoamine system, the contribution of the cholinergic system to mood disorders is becoming increasingly evident. There are two main origins of ACh in the CNS: (1) diffusely innervated projection nuclei in distal regions, including the medial septum [31], and (2) localized interneurons, represented by the striatum and NAc [32]. Human imaging studies demonstrated that enhanced cholinergic signaling triggered depression in the far past. The peripheral administration of the acetylcholinesterase (AChE) antagonist physostigmine evoked anxiety and depressive symptoms in human subjects by decreasing ACh catabolism and increasing neurotransmitter activities [33]. Rodent experiments also confirmed that increasing ACh via the acute administration of physostigmine induced anxiety-like and depressive behaviors, and prolonged treatment with the 5-hydroxytryptamine (5-HT) antidepressant fluoxetine increased AChE activity, particularly in the hippocampus [34]. These results indicate that overactive Ach signaling contributes to depression.

Two types of receptors, the metabotropic muscarinic receptors (mAChRs) and the ionotropic nicotinic receptors (nAChRs), mediate the action of Ach. The mAChRs are G protein-coupled receptors. The M1, M3, and M5 receptor subtypes couple to Gαq, and M2 and M4 couple to Gαi. The nAChRs are nonselective, excitatory cation channels that are composed of α-subunits (α2–α7) and β-subunits (β2–β4) assembled as homodimers or heterodimers [35]. Many researchers reported that nicotine improved depressive symptoms in some cases. Smoking abstinence and withdrawal from nicotine may be accompanied by depressive episodes [36]. These correlations suggest that altered nAChR signaling is involved in mood changes. Therefore, the use of nicotine from tobacco may be suggested for the treatment of depression. Extensive investigations indicated that diminished α4β2 nAChR activity alleviated the symptoms of depression [37]. Partial agonists of α4β2 nAChRs may be used alone or in combination with monoaminergic agents to alleviate depression-like behaviors in mice [38]. Notably, popular antidepressants are also α4β2 nAChR antagonists in cell-based assays [39], which demonstrates that these drugs may synergize with nAChR signaling. The antidepressant-like effects of guanfacine, a α2-norepinephrinergic receptor (α2aR) agonist, were inhibited by knockdown of α2aRs. Knockdown of the β2 nAChR subunit in the amygdala also prevented the antidepressant-like effects of guanfacine. The ablation of NE terminals blocked the anxiolytic and antidepressant-like effects of the nicotinic partial agonist cytisine. These results demonstrated that norepinephrine (NE) signaling via α2aRs and cholinergic receptors that contain the β2 nAChR subunit in amygdala neurons was essential for the modulation of depression- and anxiety-like behaviors [40]. ACh–NE interactions may be relevant for understanding the etiology of anxiety and depression and may provide new possibilities for exploiting therapeutic approaches for mood control. Other nAChRs also exert antidepressant actions. A recent study indicated that a functional nAChR containing the β4 subunit was required for the full effect of the atypical antidepressant bupropion [41]. Increasing α7 nAChR signaling combined with an increase in serotonin signaling may also enhance antidepressant efficacy [42]. The VTA receives cholinergic input from the dorsal tegmental area (DTA), which modulates phasic dopamine activity from the VTA to the NAc pathway and results in a rapid but transient increase in extracellular dopamine (DA) in the NAc. This pathway plays an important role in modulating depression-like behavior in rodents [43]. Intra-VTA administration of the cholinesterase inhibitor physostigmine or the muscarinic receptor agonist pilocarpine induces anxiety- and depression-like behaviors [44]. Co-infusion of the muscarinic M5-selective negative variant modulator VU6000181 with physostigmine in the VTA attenuated the depression- and anxiety-like behaviors induced by physostigmine alone, which reveals a critical role of M5 receptors in mediating the anxiety- and depression-like behaviors caused by increased cholinergic tone in the VTA [45]. M5 receptors anchor to the dopaminergic neuron soma in the VTA and facilitate the release of DA [46]. In contrast, the activation of M5 receptors in the striatum diminished DA release [46]. In conclusion, these findings reveal cholinergic mechanisms of depression and anxiety, particularly muscarinic M5 receptors, which may be therapeutic targets. Because the potency of AChR signaling may be very different and depends on the receptor subtype, the antagonists/inverse agonists/partial agonists of heterodimeric or homodimeric nAChRs and the antagonists of M5 mAChR with different effects may be the most promising drug complexes to treat mood disorders.

The location of cholinergic receptors in the brain also determines the effect on mood. Cholinergic blockade in several brain regions, including the amygdala, reverses depression-like phenotypes [47]. These investigations support the hypothesis that enhanced cholinergic signaling contributes to depressive symptoms. However, cholinergic signaling produces opposite effects in other brain regions. For example, reducing cholinergic transmission in the NAc induced a depressive effect in mice via the disinhibition of medium-size multi-spiny neurons [48]. Genetic, species, and sex differences further complicate the nicotinic pharmacology of depression-like behaviors [49].

More selective antagonists or inverse agonists may inhibit signaling via specific nAChR isoforms and may be more effective or better tolerated. Therefore, future clinical trials of targeted nicotinic antagonists or inverse agonists may be of interest. This also supports the potential of M5 receptors as targets for depression and anxiety.

### 3.3. Thyroid Hormones

The production and secretion of thyroid hormone (TH) are governed by the hypothalamic–pituitary–thyroid (HPT) axis. TH modulates thyrotropin-releasing hormone (TRH) and thyroid-stimulating hormone (TSH) in circulation via a negative feedback circuit. The predominant role of TH is to regulate metabolism, cell growth, and progression. TH is associated with neuropsychiatric disorders, such as schizophrenia, bipolar disorder, anxiety, and depression [50].

TH is primarily involved in later neurodevelopment, including neural migration or neuronal–glial cell differentiation, and it is implicated in adult neurogenesis [51], which occurs primarily in two regions of the brain, known as the subventricular and subgranular regions. Deficiency in adult neurogenesis is typically correlated with cognitive deficits, psychiatric disorders, and depression [52]. TH also modulates the morphology and connectivity of GABAergic neurons by governing tyrosine kinase B receptors (TrkB) during early developmental stages [53]. TH deficiency during the early developmental stage results in a decline in parvalbumin (PV)-positive neurons in rats [54], which indicates that TH is also involved in the formation of cortical circuits during early development.

Hyperthyroidism and hypothyroidism correlate with mood disorders, personality disorders, and dementia. The central effects of TH are well known from studies in hypothyroid patients [55]. Bilaterally abnormal neuronal activities in the amygdala, hippocampus, and subgenual PFC regions correlate with the severity of depression in hypothyroid patients. TH substitution eliminates these disparities and resumes normal activity in the abovementioned areas [56]. Hypothyroidism was also found in women who suffered from postpartum depression [57]. Untreated hypothyroidism correlated with depression and anxiety, and diagnosed untreated hyperthyroidism was associated only with depression [58]. Therefore, it is necessary to investigate whether TH-related treatment may alleviate depressive symptoms and anxiety in patients with hypothyroidism and hyperthyroidism. There are multiple potential reasons to explain the incomplete response to thyroid supplementation-enhanced antidepressant therapy. TH has trophic effects on astrocytes and oligodendrocytes, including the turnover, survival, and maturation of glial progenitor cells [59]. TH also synergizes with serotonin in the brain to enhance normal subgenual PFC function and diminish abnormal neural activity in the amygdala [60].

The link between thyroid dysfunction, abnormal mood, and cognitive impairment has been well documented. For overt thyroid disorders, the treatment of psychiatric symptoms with L-T4 supplementation is an excellent option. Some depressive patients are characterized by a subclinical form of hypothyroidism (SCH). A meta-analysis demonstrated that SCH was associated with depression in younger patients but not in elderly patients [61]. However, recent research indicated a positive correlation between SCH and the risk of depression in people older than 50 years [62]. Further investigations are still needed to better assess the risk of depression in patients with SCH. Overall, TH status is critical in mood disorders. However, the underlying pathogenesis has not been fully elucidated. More attention to the physiological and pathological alterations of TH and its receptors in the CNS may provide novel opportunities to identify therapeutic targets for depression.

### 3.4. Brain-Derived Peptides

#### 3.4.1. Neurotrophic Factors

Insufficient neurotrophic signaling has been recognized as a risk factor for depression over the past decades, and promoting neurotrophic signaling has been implicated in antidepressant treatment. Neurotrophic factors, including nerve growth factor (NGF), BDNF, neurotrophic factor 3 (NT-3), and neurotrophic factor 4 (NT-4) [63], exert effects by binding to their cognate tyrosine kinase (Trk) receptors. BDNF and NT-4 bind to TrkB, and NT-3 binds to TrkC and the neurotrophic factor receptor p75^NTR^.

Most studies link BDNF to mood disorders. BDNF is synthesized as a precursor protein (proBDNF), which is hydrolyzed and processed by intracellular and/or extracellular proteases to form mature BDNF (mBDNF) [64]. ProBDNF preferentially binds to p75^NTR^, but mBDNF exhibits a higher affinity for TrkB receptors [65] (Figure 2). The activation of Trk and p75^NTR^ exhibits very distinct effects. The activation of TrkB receptors enhances neuronal survival, synaptogenesis, and plasticity, and the activation of p75^NTR^ drives cell death and synaptic pruning [66,67].

The first antidepressant treatment that showed an increase in BDNF was electroconvulsive therapy [68]. Several subsequent studies demonstrated that prolonged treatment with various antidepressants increased BDNF mRNA and protein in the cerebral cortex and the hippocampus [69]. All clinically available antidepressants increased the activity of TrkB signaling in the forebrain within hours of administration [70]. A comparable increase in BDNF mRNA and TrkB phosphorylation was observed after the acute administration of the rapid-acting antidepressant ketamine [71]. These data indicate that antidepressants invariably increase BDNF signaling in the rodent forebrain [72]. The administration of BDNF in midbrain regions or the hippocampus diminished depression-like behaviors and mimicked the effects of antidepressants [73]. Deleting BDNF genes or limiting BDNF in forebrain regions blocked the effect of several different antidepressants [74]. The antidepressant efficacy of ketamine disappeared in mice completely lacking BDNF in forebrain regions [75]. However, a comparable loss of effect was not observed in heterozygous BDNF-deficient mice [76], which indicated that low levels of BDNF were sufficient to mediate an antidepressant effect of ketamine compared to classical antidepressants. However, the administration of BDNF to the NAc facilitated depression-like behavior [77], which demonstrated a network-dependent role of BDNF in mood regulation. It is emphasized that BDNF works by facilitating the network’s activity in depression-like or antidepression-like manners.

A single nucleotide polymorphism was identified in the human BDNF gene that resulted in a non-conservative mutation at codon 66 in the pre-structural domain (V66M) and led to a volume reduction in specific brain regions [78]. There is accumulating evidence that this polymorphism is associated with the pathophysiology of neuropsychiatric and neurodegenerative disorders, such as depression, anxiety, memory, and learning disorders [79]. The conformational flexibility and structural plasticity of the V66M polymorphism paved the way to understand multifaceted BDNF/TrkB signaling, including receptor binding, signaling stability, processing, and transport. The constitutive and activity-dependent activation of mature BDNF may have a profound impact on the pathophysiology of neuropsychiatric disorders.

BDNF and TrkB in the brain are downregulated in depressed patients [80]. Conversely, mRNA transcription for the neurotrophic factor receptor p75NTR was elevated in the brains of suicide victims [81]. The pan-neurotrophic factor receptor p75NTR is almost exclusively found in adult cholinergic neurons in the basal forebrain. NMDA-induced long-term inhibition (LTD) was impaired in the hippocampus of juvenile *p75NTR*^−/−^ mice but was normal in adult *p75NTR^−/−^* mice [82]. One possible explanation is that p75^NTR^ deficiency contributes to alterations in hippocampal cholinergic signaling, which may be involved in the modulation of stress-activated hippocampal LTD and stress-induced anxiety-like behaviors.

The neuronal plasticity enhanced by neurotrophic factors is converted into an antidepressant response [83]. Antidepressants foster neuronal plasticity in several dimensions [84]. First, antidepressants increase neurogenesis. An increase in neurogenesis occurs during chronic antidepressant treatment, and the process depends on BDNF signaling. Second, antidepressants increase axonal elongation, dendritic sprouting, and plasticity-related proteins. However, whether BDNF or TrkB signaling mediates these effects of antidepressants is not known. Finally, chronic antidepressant treatment facilitates synaptogenesis and increases synapse intensity. These data indicate a strong correlation, and a causal relationship in some cases, between BDNF signaling, neuronal plasticity, and antidepressant effects. Therefore, BDNF signaling may be a valuable target for the screening of novel anti-depressive drugs.

#### 3.4.2. Sortilin and TWIK-Related K^+^ Channel 1 (TREK-1)

As a cellular receptor or neurotrophic factor co-receptor, sortilin dysregulation contributes to the progression of a variety of diseases. Sortilin, also known as neurotensin receptor 3 (NTSR3), is a member of the vesicular protein sorting 10 protein (Vps10p) domain family, that was initially identified in the human brain but is also expressed in other tissues [85]. Sortilin is implicated in cell survival, signal transduction, and protein transport, such as transferring neurotrophic factors to the plasma membrane or lysosomes via sorting or endocytic pathways [85]. Sortilin mediates the endocytosis of adiponectin and nerve growth factor precursors and acts as a co-receptor for p75^NTR^ on the cell membrane to trigger neural death [86]. It forms a heterodimeric complex with p75^NTR^ at the cell surface to transduce pro-neurotrophic factor-cell death signals, and it works together with Trks to enhance neurotrophic factor signaling by facilitating cis-axis transport. In response to various stimuli, the extracellular structural domain of sortilin is cleaved and forms approximately 5–10% of soluble sortilin molecules (sNTSR3) [85]. sNTSR3 binding to a receptor (not yet identified) stimulates Akt, Src, and Fak phosphorylation to modulate intracellular pathways involved in cell survival [87]. sNTSR3 suppresses the conversion of proBDNF to mBDNF and has apoptotic properties that prevent neuronal damage from proBDNF (Figure 2). Residual intrinsic sortilin is a ligand-binding site that transduces signals or mediates endocytosis [85]. High sortilin levels were found in depressed patients [88], and changes in serum sortilin and sortilin mRNA expression were observed after antidepressive treatment [89]. These findings suggest that sortilin is a potential biomarker for depression. However, a recent study based on a relatively small sampling group indicated that neither marked changes in serum sortilin nor a correlation between serum sortilin and depression scores was observed after 12 weeks of antidepressant treatment [90]. Nevertheless, a positive correlation between sortilin and BDNF was reported in depressive patients [88], which suggests that sortilin plays an essential role in modulating BDNF activity. Notably, current findings indicated that antidepressant treatment in mice did not alter sortilin in the hippocampus or the cortex [88]. Therefore, the elevated sortilin in the serum of depressive patients may be attributed to depression rather than treatment. Taken together, sortilin function and expression are fine-tuned to facilitate normal physiological procedures, and its imbalance may contribute to the progression of many human diseases.

TREK-1 is a new target for antidepressants (Figure 2). This mechanosensitive channel is abundant in GABAergic neurons in the caudate nuclei [91] and glutamatergic neurons in the hippocampus [92]. At clinical concentrations, fluoxetine and its active metabolite norfluoxetine blocked TREK-1 channels in a concentration-dependent manner [93]. Recent research demonstrated that other SSRIs, such as paroxetine and citalopram, were also potent blockers of TREK-1 channels in HEK-293 cells and HT-22 neuronal cells [94].

Knockout of *kcnk2*, which is the gene encoding the TREK-1 channel, also revealed the involvement of the TREK-1 channel in depression. Knockout mice exhibited a depression-resistant phenotype compared to wild-type control mice in behavioral assays [95]. Deletion of the *kcnk2* gene enhanced the firing of 5-HT neurons in the dorsal raphe nucleus (DRN) [96]. It is important to screen for new specific TREK-1 inhibitors with antidepressant properties using hTREK-1/HEK cell lines stably expressing human TREK-1 channels [97]. Star*D studies in humans identified an association between single nucleotide polymorphisms in the TREK-1 locus and resistance to multiple antidepressants [98]. Taken together, these investigations in humans reinforce the view that TREK-1 is a pivotal target of depression treatment, and the identification of selective blockers of TREK-1 may contribute to the emergence of a new generation of antidepressants.

TREK-1 channel expression and function were altered in mice with sortilin deficiency [99], which further supports TREK-1 as a crucial target for the treatment of depression. Mice with genetic deletion of sortilin were linked to TREK-1 dysfunction and exhibited a reduction in depression-like behavior and increased neuronal activity in the DRN [99]. *Sort1^−/−^* mice showed reduced TREK-1 expression compared to WT mice, which indicates a critical role of sortilin in the correct sorting of TREK-1 channels. A previous study also demonstrated the importance of sortilin in TREK-1 transportation. Overall, these studies support the involvement of TREK-1 and NTSR3 in the physiopathology of depression.

#### 3.4.3. Sortilin-Derived Propeptide (PE) and Spadin

Post-translational cleavage of the sortilin precursor by the protein convertase furin yields mature NTSR3 and releases a 44-amino-acid peptide termed PE [100]. The peptide sequence required for the binding of PE to NTSR3 was identified as Gln1-Arg28. Spadin is a 17-amino-acid peptide (Figure 2) that contains the major fragment of PE and is capable of NTSR3 binding [101]. This peptide is a potent and selective antagonist of TREK-1 [101] and a crucial target for the treatment of depression [102].

Several studies in rodents identified spadin as an endogenous and rapid-acting antidepressant [101,103]. Spadin enhanced 5-HT neurotransmission by blocking TREK-1 [101] and produced a similar effect as that observed in *kcnk2^−/−^* mice [104]. Subchronic treatment with spadin increased cAMP response element binding protein (CREB) activation and hippocampal neurogenesis. This effect is striking because it occurred only after four days of treatment, and SSRIs need 3–4 weeks to increase hippocampal neurogenesis [101]. Spadin also increased the mRNA and protein expression of synaptogenic markers, such as PSD-95 and synaptophysin [105]. Spadin also dramatically increased the number of mature spines [105]. The major advantage of spadin over conventional antidepressants may be the absence of side effects. Spadin did not elicit the cardiac dysfunction associated with TREK-1 inhibition [106]. In summary, these findings demonstrate that spadin is a natural endogenous antidepressant with a rapid onset of effect. Given these specific properties, spadin presents a new concept to address the treatment of depression. Spadin is also the first identified TREK-1 channel blocker. Reliable methods to detect PE and spadin are essential for the future adoption of spadin as a marker of depression in depressive patients.

A recent cohort study found a correlation between serum PE in healthy individuals and MDD patients (with or without medication) [107], and MDD patients had markedly lower serum PE concentrations. Notably, a partial recovery of PE was observed after 12 weeks of antidepressant treatment. Although sortilin is overexpressed in MDD patients, a new report concluded that sortilin was not a biomarker in MDD patients after antidepressant treatment [90]. How do PE levels decrease when sortilin expression and release are upregulated in MDD patients? Further thorough clinical studies, including larger cohorts of MDD patients with or without antidepressant treatment, are essential to validate whether PE and sortilin may serve as biomarkers for MDD. Alternatively, progress in developing specific antibodies that recognize short peptide sequences with greater sensitivity would guide the development of new biomarkers of MDD.

Shortened spadin analogs, such as mini-spadin, recently exhibited better efficacy and in vivo stability than spadin [97]. The beneficial effects may be due to its biphasic action on TREK-1 channels. Low doses of mini-spadin evoked TREK-1, and the activation of TREK-1 in neurons hyperpolarized synaptic terminals, decreased Ca^2+^ influx and glutamate release, and produced postsynaptic hyperpolarization. Mini-spadin dramatically reduced depression-like behavior in the chronic phase of stroke [108]. Can sortilin-derived propeptides be validated as viable drug targets for the treatment of MDD? Future studies are likely to reveal new perspectives on the significance of sortilin or sortilin-derived propeptides and their relatives in the intervention of neuropsychiatric diseases.

#### 3.4.4. Opioid Peptides

The endogenous kappa opioid system affects behaviors associated with motivation and emotional states in animal models of depression and anxiety [109]. These findings evoked interest in exploring selective antagonists of kappa opioid receptors (KORs) as potential pharmacotherapies for mood disorders. Behavioral investigations indicated that KOR antagonists exerted antidepressant and anxiolytic effects in animals [110]. Whereas antagonist activity has a generally slow onset and is extremely sustained in vivo [111], the exploitation of short-acting KOR antagonists with decent blood–brain barrier (BBB) permeability and bioavailability represents exciting progress toward opioid identification as potential antidepressants, anxiolytics, and anti-addictive drugs.

The nociceptin opioid peptide (NOP) receptor and its endogenous ligand nociceptin/orphanin FQ (N/OFQ) is the fourth member of the opioid receptor and opioid peptide family [112]. Knockout of the NOP receptor gene generated anti-depression-like behavior in rodents [113]. Agonists exacerbated depressive behaviors [114]. NOP receptor antagonists revived stress-induced changes in monoamine levels in the rodent brain [113]. High concentrations of N/OFQ reduced the cellular release of 5-HT [115]. To complicate matters, NOP receptor agonists prevented the antidepressant effects of nortriptyline and fluoxetine, but not ketamine [116]. This finding suggests that ketamine exerts antidepressant activity via a different mechanism than SSRIs. Plasma levels of N/OFQ were considerably higher in patients with postpartum depression, bipolar depression, and MDD compared to healthy subjects [117]. These findings indicated that NOP receptor antagonism may be an antidepressant in MDD patients. Recent work demonstrated that LY2940094, a novel, potent, and selective oral antagonist of the NOP receptor, was a safe and well-tolerated antidepressant in rodent models and patients with MDD [118]. The blockade of NOP receptor signaling is a promising strategy for the treatment of MDD. The exploitation of nociceptin/orphanin FQ-based antagonists presents a new strategy to cope with depression.

#### 3.4.5. Oxytocin and Arginine Vasopressin

Oxytocin (OXT) and arginine vasopressin (AVP) have become promising targets for psychiatric disease treatment. Anxiety, stress, and positive or negative stimuli drive the synthesis and release of OXT and AVP from neurons in the hypothalamic supraoptic nucleus (SON), paraventricular nucleus (PVN), or limbic system regions [119]. OXT and AVP are also critical regulators of anxiety and depression-like behaviors [120].

Synthetic OXT showed an antidepressant-like effect following central or peripheral administration in rodents [121]. Overall, perinatal OXT negatively correlated with depressive symptoms. Eight of the 12 studies focusing on endogenous oxytocin revealed a negative correlation between plasma OXT and depressive symptoms [122]. Notably, some of the effects of SSRIs were mediated by OXT [123].

The overexpression of AVP in the PVN induced depression-like behavior in a rat model of anxiety, which was normalized by long-term treatment with the antidepressant paroxetine [124]. Similarly, the number of neurons expressing AVP was increased in the PVN of depressed patients [125]. Therefore, shifting neuropeptide homeostasis to OXT by suppressing brain AVP may be beneficial in the treatment of depression. In addition to the appropriate stimulation of the endogenous system, combined psychopharmacological treatment with OXTR agonists and AVPR antagonists may synergistically improve psychopathic behavior.

### 3.5. Non-Brain-Derived Peptides

#### 3.5.1. Adiponectin and Leptin

Adiponectin is a 244-amino-acid polypeptide protein generated from adipocytes alone [126]. Its transcription is governed by the peroxisome proliferator-activated receptor (PPAR) [127]. Adiponectin promotes neurogenesis in the hippocampus via the adiponectin receptor AdipoR1 [128], which directly affects synaptic function via AdipoR2 [129]. A reduction in adiponectin suppressed neurogenesis in the dentate gyrus (DG) of adult male mice, and intravenous administration of adiponectin promoted neurogenesis [130].

Plasma adiponectin levels were higher in depressed patients who received antidepressants for longer than 12 months [131]. Studies on model mice suggested that the alleviating effect of physical exercise on depressive symptoms may be mediated by adiponectin, which facilitated hippocampal neurogenesis [128]. The antidepressant activity of PPARγ agonists may be due to elevated adiponectin expression. Notably, the PPARγ agonist rosiglitazone exhibited antidepressant and anxiolytic effects in WT control mice, and this effect was absent in mice lacking adiponectin [132]. The adiponectin-notch pathway is involved in the pathogenesis of depression-related cognitive dysfunction and impaired hippocampal neurogenesis [133].

There is emerging evidence that adiponectin and adiponectin receptors are critical targets in translational research to identify novel therapeutic and/or preventive strategies for various CNS disorders, such as anxiety and/or depressive disorders. However, detailed studies on brain adiponectin signaling and its mechanism are necessary to identify specific agonists or antagonists of adiponectin receptors that may be used as a therapeutic strategy for CNS disorders in future clinical trials.

The accumulating evidence from preclinical studies unequivocally demonstrated that leptin had antidepressant effects. Leptin is a hormone predominantly made by adipose cells and enterocytes in the small intestine that helps to regulate energy balance. The direct activation of leptin receptors expressed in the hippocampus and amygdala results in enhanced neurogenesis and neuroplasticity in hippocampal and cortical structures and altered HPA axis activity [134,135]. Clinical investigations based on a small group of MDD patients revealed that higher leptin levels correlated with atypical MDD. This correlation was strengthened in obesity [136], which reinforces the hypothesis that leptin resistance is involved in depression. The relationship between leptin, HPA axis activity, and anxiety in suicide attempters was reported recently [137]. Cerebrospinal fluid leptin/body mass index significantly and negatively correlated with anxiety and serum cortisol before and after dexamethasone suppression tests. In initially non-depressed subjects, those with high levels of leptin and abdominal obesity were at higher risk of developing clinically relevant depressive symptoms nine years later. Current research offers new hope to patients with depression and anxiety by directly targeting leptin-expressing neurons to alter pathological changes. However, further investigations are warranted to determine whether genetic polymorphisms in leptin signaling (particularly the leptin receptor) are relevant to atypical MDD and elucidate the mechanisms underlying the overall function of leptin and the receptor pathway. In the long term, developing treatments that effectively alleviate leptin resistance may benefit patients with atypical depression characterized by obesity-related metabolic alterations.

#### 3.5.2. Ghrelin

Ghrelin is a 28-amino-acid orexigenic hormone that is synthesized in the stomach and binds to the growth hormone secretory receptor (GHSR) to modulate the growth-promoting secretion of pituitary hormones [138]. There are two forms of ghrelin circulating in blood plasma, acylated and nonacylated ghrelin, both of which are capable of crossing the BBB. However, only acylated ghrelin binds to GHSR [139]. GHSR is extensively present in the brain [140], including the arcuate nucleus (ARC), amygdala, hippocampus, NAc, and VTA. Recent studies highlighted that ghrelin and GHSR1a play complex roles in the regulation of a diverse number of brain functions, including hunger and metabolism, learning and memory, reward and addiction, motivation, stress responses, anxiety, and depression [141,142]. Ghrelin seems to play a dual role in anxiety and depression depending on the contextual and physiological states.

Early work demonstrated that ghrelin alleviated anxiety and depression-like behaviors [143,144,145]. The central administration of ghrelin increased dopamine release by activating ghrelin receptors, and systemic treatment with the ghrelin receptor antagonist JMV2959 prevented dopamine release in the NAc [146]. Depression is accompanied by a decrease in BDNF and a concomitant decrease in CREB, followed by behavioral changes and an increase in depressive states [147]. Ghrelin stimulates the GHSR, which activates VTA dopaminergic neurons and the projected dopaminergic pathway, increases BDNF with an eventual increase in CREB, and alleviates depressive symptoms [148]. A recent investigation indicated that ghrelin modulated CREB in damaged nerves and was involved in neuronal morphogenesis [149]. The administration of ghrelin resulted in hippocampal dendritic spine remodeling and increased BDNF mRNA [150]. The antidepressant effect of ghrelin may also be associated with its enhancement of synaptic plasticity in VTA neuronal circuits [151]. GHSR knockout mice exhibited dramatically increased depression-like behaviors in response to chronic social defeat stress [152]. Three-year follow-up research revealed a negative correlation between ghrelin and chronic depression in elderly male patients [153]. Central ghrelin administration alleviated depression-like behaviors triggered by chronic unpredictable mild stress [145]. Depressive symptoms resulting from catecholamine depletion in healthy individuals also correlated with a decrease in plasma ghrelin [154]. All of these findings suggest a potential antidepressant effect of ghrelin.

Paradoxically, ghrelin also promoted anxiety and depression [143,155,156,157]. We previously reported that GHS-R1a deficiency alleviated depression-related behaviors after chronic social defeat stress [152]. Elevated ghrelin was also detected in patients suffering from MDD [158]. A recent study suggested that, besides a “hunger” hormone, acyl-ghrelin was also a persistent biomarker for chronic stress exposure [159]. Acyl-ghrelin levels remained elevated in C57BL/6 mice after chronic social defeat stress (CSDS) [160], in rats after chronic immobilization stress [157,161], and in mice after chronic unpredictable mild stress [145]. A prolonged increase in the circulating level of acyl-ghrelin was observed in adult or adolescent rodents and vulnerable adolescent humans exposed to a chronic severe stressor [157,159,161]. Although the elevation of circulating acyl-ghrelin induced by chronic, psychological stress is always accompanied by exacerbated anxiety- and depression-like behaviors, the reason for the chronic stress induction of ghrelin release and the contribution of ghrelin to stress responses and mood regulation remains uncertain.

Many other peptides, including relaxin, orexin, and apelin, also play roles in antidepressant therapy. These receptors form structural heterodimers with 5-HT receptors [162] or exhibit functional crosstalk with the HPA axis [163]. Compounds targeting these endogenous peptides and their receptors may address the limitations of present antidepressants in the future.

### 3.6. Cannabinoids

The endocannabinoid system (ECS) is a ubiquitous lipid signaling system that participates in a variety of intracellular signaling pathways. Endocannabinoids bind to cannabinoid receptors (CBRs) to regulate many physiological functions, modulate crosstalk between different neurotransmitter systems, and play critical roles in the control of behaviors [164]. There are three types of CBRs, the G protein-coupled receptors CB1 and CB2, ligand-sensitive ion channels (e.g., transient receptor potential vanilloid 1 (TRPV1)), and nuclear receptors (e.g., PPAR) [165]. *N*-arachidonoylethanolamine (AEA) and 2-arachidonoylglycerol (2-AG), derived from polyunsaturated fatty acids, are endogenous ligands for cannabinoid receptors [165]. 2-AG is primarily involved in CB1-dependent retrograde signaling and is an intermediate metabolite of lipid synthesis. Monoacylglycerol lipase (MAGL) is present predominantly at the presynaptic terminal to breakdown 2-AG [164].

There is a large amount of evidence from clinical studies demonstrating the pivotal role of ECS in depression. Untreated MDD patients had low basal serum concentrations of AEA and 2-AG, and exposure to stressful environments markedly increased 2-AG concentrations without altering AEA [166]. Another clinical study reported that elevated 2-AG concentrations significantly correlated with antidepressant treatment with SSRIs [167]. Electroconvulsive therapy significantly increased AEA and 2-AG levels in the cerebrospinal fluid of MDD patients [168]. A relationship between genetic variants in cannabinoid receptor type 1 and type 2 genes (*CNR*; *CNR1* and *CNR2*) and susceptibility to depression was also reported. A recent meta-analysis indicated that *CNR1* rs1049353 or AAT triplet repeat sequence polymorphisms were not relevant for depression susceptibility, but *CNR2* rs2501432 gene polymorphisms may be strongly correlated with depression [169].

CB1R may be a novel and promising target for depression treatment. Single nucleotide polymorphisms in the *CNR1* gene were associated with depression and other mood disorders [170]. Increased desperate behavior in *CNR1^−/−^* mice closely correlated with downregulated BDNF in the hippocampus. Local administration of BDNF in the hippocampus reversed the depression-like phenotype [171]. The enhancement of endogenous cannabinoid signaling is a promising pharmacological strategy for the treatment of stress-related disorders, e.g., anxiety or depression. A significant reduction in depression-like behaviors was found following administration of the FAAH inhibitors URB597 [172] or PF3845 [173]. MAGL suppression by JZL194 administration had similar antidepressant effects. Notably, JZL194-mediated effects may be correlated with enhanced adult neurogenesis and long-term synaptic plasticity in the dentate gyrus [174]. JZL195, a dual inhibitor of FAAH and MAGL, increased endogenous cannabinoid and BDNF levels in the rat ventral striatum and decreased depressive-like behaviors [175]. Fluoxetine altered the concentrations of different ECS components in the basal state and under depressive conditions [176]. The combination of sub-effective doses of fluoxetine with sub-effective doses of AEA, AM404, or URB597 facilitated the antidepressant effect of these compounds [177].

Cannabidiol (CBD) is a natural component of cannabis resin flowers that has antipsychotic and anxiolytic effects. CBD reduced depressive symptoms in a mouse model [178]. The mechanism may involve the activation of the 5-HT1A receptors. It was also hypothesized that CBD increased BDNF to alleviate depression [179]. The short-term antidepressant effect of CBD was associated with increased expression of synaptophysin, PSD95, and BDNF in the mPFC and the hippocampus [180]. Chronic CBD treatment also increased BDNF levels in the amygdala [180].

The wide distribution of the ECS system and the pathophysiological role of ECS signaling in many brain disorders provide good opportunities for the development of novel cannabinoids, cannabis generic, and cannabis-based drugs to treat psychiatric disorders, such as depression. Further studies are necessary to reveal the molecular mechanisms underlying ECS signaling in mood control.

### 3.7. Endogenous Digitalis-Like Compounds

Neurons and/or glial cells synthesize bioactive steroids from cholesterol in much the same way as peripheral steroid production. These brain-derived steroids (called neurosteroids) regulate a huge range of behavioral and metabolic activities [181].

Digitalis and digitalis-like compounds (DLCs) have been identified in human tissues in recent decades and are considered a new family of steroid hormones. The binding of digitalis to specific sites on Na^+^, K^+^-ATPase results in the inhibition of ATP hydrolysis and ion transport [182]. DLC is involved in the regulation of major physiological parameters at the systemic level, including water and salt homeostasis, cardiac contractility and rhythm, systemic blood pressure, and behavior [183]. A vast number of publications have indicated the presence of DLC in the brain, which may work as a neurotransmitter or neuromodulator [184,185].

Brain DLC system activity has been implicated in the etiology of depression. The neutralization of endogenous DLC by the acute administration of anti-ouabain antibodies in SD rats triggered antidepressant behaviors and altered the levels of catecholamines and their metabolites in brain regions associated with depression [184,186]. For example, the administration of anti-ouabain antibodies resulted in a dramatic decline in the levels of DA and its metabolites in the hippocampus. Notably, 5-HIAA and the 5-HIAA/5-HT ratio in the hippocampus were also markedly decreased after the application of ouabain antibodies [184]. These findings are consistent with the idea that malfunction in the DLC system is associated with the etiology of depression and indicate that the endogenous DLC system may be a target for antidepressants. DLC was substantially higher in the brains of patients with bipolar disorder than in normal subjects. An association between genetic polymorphisms of the Na^+^, K^+^-ATPase α isoform and bipolar disorder was also identified [187].

Endogenous inhibitors of Na+, K^+^-ATPase that are functionally similar to DLC have been gradually identified, such as marinobufagenin and telocinobufagin [188]. These steroids competitively bind to ouabain-specific sites rather than alter the activity of ATP hydrolase. Substitution of the ATP binding site eliminated the transport function of the enzyme but did not alter ouabain binding [189]. Future investigations should investigate the relevance of these receptor-ligand interactions under various physiological conditions and the corresponding role of Na^+^, K^+^-ATPase inhibitors or activators in mood regulation. These findings will provide compelling evidence for the ongoing exploitation of novel antidepressants.

## 4. Concluding Remarks

Presently available antidepressants, such as SSRIs and ketamine, either have unpleasant side effects, e.g., suicide, addictive properties, or the ability to induce schizophrenia. Therefore, developing new, fast-onset antidepressants without these drawbacks is still an important goal for neuropharmacological research. In this review, we summarized recent studies demonstrating antidepressant effects by the manipulation of interactions between heterogeneous compounds and their receptors. Although not directly targeting the serotonin transporter (SERT), all of these manipulations showed facilitation in BDNF release, serotonin transmission or adult neurogenesis, suggesting the potential of these endogenous compounds and their receptors as new biomarker or targets for antidepressant treatment (Figure 3). However, in-depth studies are still required to understand how these heterogeneous ligands and receptors affect monoamine neurotransmission, neurogenesis, neural connectivity, synaptic plasticity, and receptor–receptor interactions. How to design an antidepressant with high selectivity on acting targets, such as receptor subtype and receptor location, is still a key and a big challenge to achieve antidepressant efficiency while alleviating side effects. Strikingly, a very recent study discovered that blocking the SERT–neuronal nitric oxide synthase (nNOS) interaction by small molecules called SNBIs specifically enhanced the activity of DRN serotonergic neurons and promoted serotonin release from DRN into the mPFC, thereby producing a fast-onset antidepressant effect [190]. We hope that our review will shed light on the importance of the heterogeneous ligand and receptor interaction for the pathogenesis of depression, eventually helping to open a new avenue for developing therapeutics for mood disorders.

## Figures and Tables

**Figure 1 ijms-23-13776-f001:**
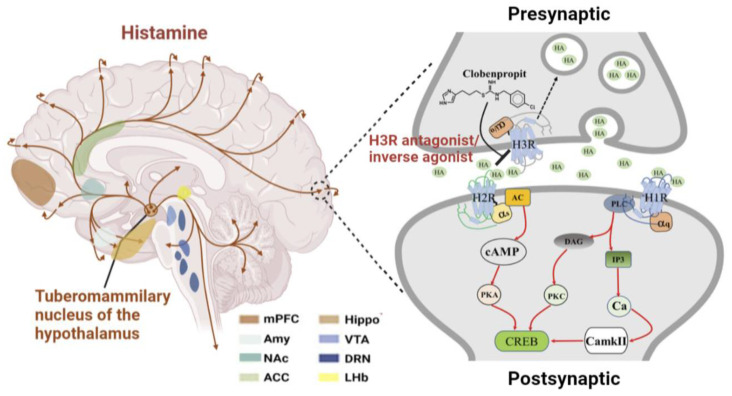
Schematic diagram illustrating the working mechanism of the H3R antagonist/reverse agonist in depression. HA is primarily derived from histaminergic neurons located solely in the tuberomammillary nucleus in the posterior hypothalamus and which project axons throughout the brain including critical brain regions associated with depression. Presynaptic inhibitory H3R regulates HA secretion by negative feedback (dotted black line). H3R antagonists/reverse agonists, such as Clobenpropit, exhibit an antidepressant effect by inhibiting H3R and thus increasing HA release (solid black line). Postsynaptic H1R and H2R modulate intracellular signaling pathways (solid red line). Abbreviations: HA, histamine.

**Figure 2 ijms-23-13776-f002:**
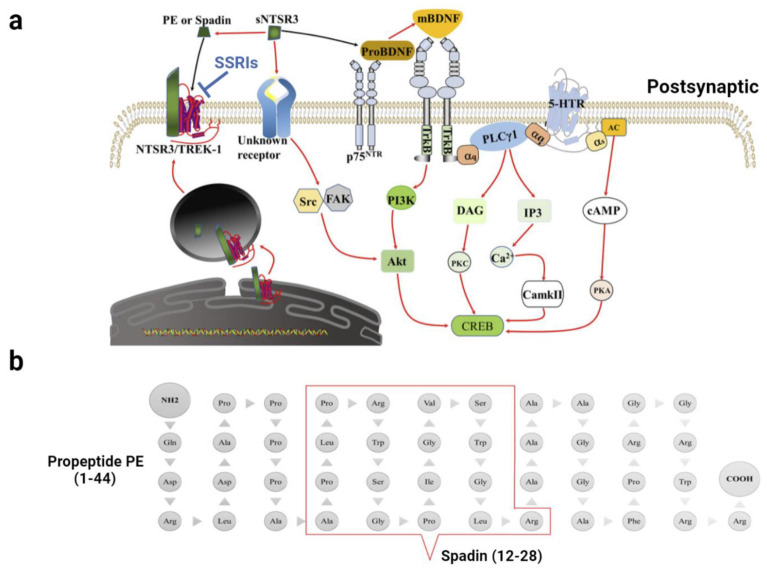
Schematic diagram illustrating the working mechanism of brain-derived peptides. (**a**) The working mechanism and signaling pathways of brain-derived peptides. proBDNF preferentially binds to p75^NTR^ to drive cell death and synaptic pruning, while mBDNF has a higher affinity for the TrkB receptors to enhance neuronal survival and synaptogenesis. TrkB simultaneously interacts with serotonin receptors. TREK-1 and NTSR3 from the endoplasmic reticulum are co-localized on the cell membrane after shearing maturation. They functionally interact with each other. The extracellular structural domain of NTSR3 is cleaved to form sNTSR3. sNTSR3 can activate downstream signal pathways via unknown receptors (solid red line); meanwhile, it inhibits BDNF maturation (solid black line). sNTSR3 can also be sheared to generate PE or Spadin (solid red line), which blocks the TREK-1 channel (solid black line) and exerts an antidepressant effect. SSRIs are potent blockers of TREK-1 channels. (**b**) Comparison of the peptide sequence of PE and spadin. PE is a propeptide containing 44 amino acids, and spadin has only the peptide sequence at Ala12–Arg28 in PE. Abbreviations: proBDNF, precursor BDNF; mBDNF, mature BDNF; TrkB, tyrosine kinase B; NTSR3, neurotensin receptor 3, or sortilin; sNTSR3, soluble NTSR3; TREK-1, TWIK-related K^+^ Channel 1.

**Figure 3 ijms-23-13776-f003:**
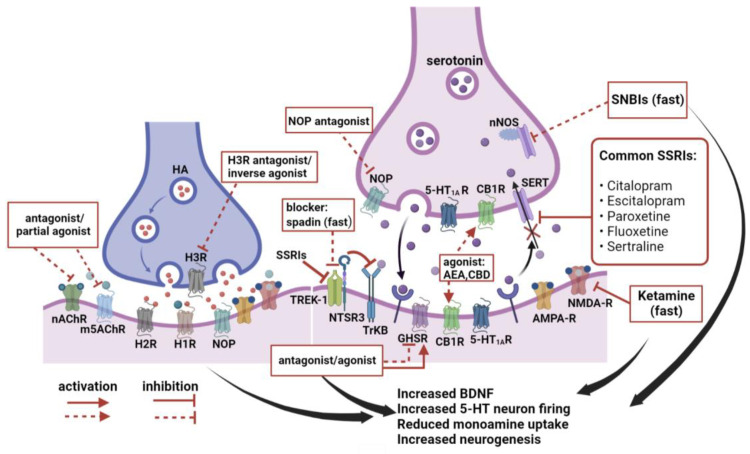
Schematic diagram summarizing potential targets for antidepressants. SSRIs and ketamine are currently available antidepressants; their pharmacological targets are indicated by the solid red line. Other potential targets, including nAchR, m5AchR, TREK-1, GHSR, CB1R, and presynaptic H3R, NOP, and SERT-nNOS, are indicated by the dotted red line. Compounds with antidepressant effects are indicated as antagonists, agonists or blockers. Abbreviations: HA, histamine; NOP, nociceptin opioid peptide receptor; TREK-1, TWIK-related K^+^ Channel 1; NTSR3, neurotensin receptor 3, also known as sortilin; GHSR, growth hormone secretory receptor; CB1R, cannabinoid receptor 1; CBD, cannabidiol; AEA, N-arachidonoylethanolamine; SERT, serotonin transporter; nNOS, neuronal nitric oxide synthase; SNBIs, SERT-nNOS binding inhibitors; SSRIs, selective serotonin reuptake inhibitors.

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
