# Peer review of "Potential of Heterogeneous Compounds as Antidepressants: A Narrative Review"

_ijms, 2022, doi:10.3390/ijms232213776_

Round 1

Reviewer 1 Report

Publication in the form of a review is done correctly. The topic is topical. The amount of literature viewed is adequate.

The subject of depression and using drugs is still valid, but it is complicated (mainly in the form of a review) to bring something new to this topic. The attempt to indicate the potential of heterogeneous compounds as antidepressants is bold but very hard to make. It is complex to implement scientific undertaking properly.

However, the publication has some potential - it can be considered for publication after the changes indicated in the comments.

Remarks:

1. Criteria for inclusion and exclusion of works from the Review were not given

2. A significant part of the work concerns research from China.

3. In this type of publication, the balance between papers from different regions (Europe, USA, Africa, Asia, etc.) should be kept.

4. Provide the key based on which the Review was done (not to be confused with meta-analysis or systematic Review)

5. Concluding remarks requires redrafting, clearly indicating the authors' opinion on the subject of the Review.

6. The current form of Concluding remarks - it brings nothing to science in the indicated subject area.

7. Keywords: add a tag: Review

Reviewer 2 Report

This is a very well-structured review with enough depth to be of interest to a wide audience in the area of neuropsychiatry. This reviewer has no further comments except to request the correction of some specific grammatical errors that will be evident after a thorough checkup of the manuscript.

It seems to me, however, that this review article would be much improved if one or more final figures were included summarizing the possible mechanisms of action and interaction of the drugs mentioned as well as the possible crosstalk between the neurotransmission systems analyzed.
